# Effect of Mechanical Damage in Green-Making Process on Aroma of Rougui Tea

**DOI:** 10.3390/foods13091315

**Published:** 2024-04-25

**Authors:** Fuming Lin, Huini Wu, Zhaolong Li, Yan Huang, Xiying Lin, Chenxi Gao, Zhihui Wang, Wenquan Yu, Weijiang Sun

**Affiliations:** 1College of Horticulture, Fujian Agriculture and Forestry University, Fuzhou 350002, China; linfm529@126.com (F.L.); 5220330098@fafu.edu.cn (H.W.); gaochenxii@163.com (C.G.); 2210311001@fafu.edu.cn (Z.W.); 2Anxi College of Tea Science, Fujian Agriculture and Forestry University, Quanzhou 362406, China; yanhuang@fafu.edu.cn; 3Institute of Animal Husbandry and Veterlnary Medicine, Fujian Academy of Agricultural Sciences, Fuzhou 350003, China; lizhaolong@faas.cn; 4Fuding Tea Technology Promotion Station, Ningde 355200, China; linxiying168@163.com; 5Tea Research Institute, Fujian Academy of Agricultural Sciences, Fuzhou 350003, China

**Keywords:** Wuyi Rock Tea, transcriptomes, proteomics, volatile organic compounds, green-making process, gas chromatography

## Abstract

Rougui Tea (RGT) is a typical Wuyi Rock Tea (WRT) that is favored by consumers for its rich taste and varied aroma. The aroma of RGT is greatly affected by the process of green-making, but its mechanism is not clear. Therefore, in this study, fresh leaves of RGT in spring were picked, and green-making (including shaking and spreading) and spreading (unshaken) were, respectively, applied after sun withering. Then, they were analyzed by GC-TOF-MS, which showed that the abundance of volatile compounds with flowery and fruity aromas, such as nerolidol, jasmine lactone, jasmone, indole, hexyl hexanoate, (E)-3-hexenyl butyrate and 1-hexyl acetate, in green-making leaves, was significantly higher than that in spreading leaves. Transcriptomic and proteomic studies showed that long-term mechanical injury and dehydration could activate the upregulated expression of genes related to the formation pathways of the aroma, but the regulation of protein expression was not completely consistent. Mechanical injury in the process of green-making was more conducive to the positive regulation of the allene oxide synthase (AOS) branch of the α-linolenic acid metabolism pathway, followed by the mevalonate (MVA) pathway of terpenoid backbone biosynthesis, thus promoting the synthesis of jasmonic acid derivatives and sesquiterpene products. Protein interaction analysis revealed that the key proteins of the synthesis pathway of jasmonic acid derivatives were acyl-CoA oxidase (ACX), enoyl-CoA hydratase (MFP2), OPC-8:0 CoA ligase 1 (OPCL1) and so on. This study provides a theoretical basis for the further explanation of the formation mechanism of the aroma substances in WRT during the manufacturing process.

## 1. Introduction

Wuyi Rock Tea (WRT) is an oolong tea known for its exceptional “rock flavor”, mainly produced in Wuyi Mountain, Fujian Province, China. As a quintessential Wuyi Rock Tea (WRT), Rougui Tea (RGT) is favored by the public because of its natural flowery and fruity aroma; this is attributed to the tea varieties and the manufacturing process, of which the manufacturing process is the main reason for the formation of RGT’s unique aroma. The main manufacturing process of RGT includes plucking, withering, green-making (turnover), rolling, roasting and so on. During the manufacturing process of RGT, the main volatile substances produced include alcohols, alkenes and esters, in which the content of alcohols gradually decreases, and the alkenes and esters show wave changes, which are obviously increased during green-making, and a large number of volatile compounds containing nitrogen are formed after baking [1,2]. RGT produces a large number of alcohols and aldehydes during fermentation, such as linalool, indole, nerol, (Z)-3-hexen-1-ol and (Z)-3-hexenyl acetate [3,4,5]. Other WRTs produce similar volatile substances, such as terpene, aldehyde and ester compounds, during the manufacturing process, and green-making has a significant effect on it, finally promoting the formation of flowery or fruity aroma characteristics [6,7,8]. During the manufacturing process of WRT, the volatile content that mainly contributes to the grassy odor decreases, while the volatile content that mainly contributes to the flowery and fruity aroma increases [9].

After picking the fresh leaves of RGT, they are withered in sunlight for a short time and then subjected to green-making and the subsequent manufacturing process, which is more conducive to producing a mellow taste and flowery aroma. Sun withering can increase the gene expression of ubiquinone, other terpenoid quinones’ biosynthesis, pyruvate metabolism and the starch and sucrose metabolism pathway, which increases the content of terpenoids, organic acids and lipids, thus enhancing the mellow taste and flowery aroma of the tea [10]. Different withering degrees also affect the quality of RGT. A previous study found that moderate withering (a withering time of 30 min) could increase the content of terpenoids, organic acids and lipids in the leaves and improve the quality of the RGT [11]. It can be seen that withering affects RGT’s quality, but the effect of green-making on RGT’s quality is the most significant. Studies have shown that the biosynthesis and release of volatile organic compounds (VOCs) in plants are related to self-defense in the face of biotic and abiotic stresses [12]. Green-making causes the continuous mechanical injury and dehydration of the fresh tea leaves, which induces the transformation and release of volatile organic compounds under traumatic stress. Biotic stress and abiotic stress during the tea manufacturing process promote the formation of various aroma compounds, which has become a major research topic in recent years [13]. Quantitative real-time PCR (qRT-PCR) and transcriptome data showed that many genes related to the formation of flowery and fruity aromas were activated during green-making [14,15,16]. The content of indole, jasmine lactone and (E)-nerolidol in fresh tea leaves during green-making was positively related to the expression of the CsTSB2, CsTSA, CsLOX1 and CsNES genes [17,18,19]. Continuous mechanical injury during the manufacturing process of RGT can upregulate the expression of CsMYC2, the key transcription factor in jasmonic acid signal transduction; increase the content of jasmonic acid; and promote the formation of aroma compounds [20]. Aliphatic alcohols and esters with a natural flowery and fruity aroma that can be formed during green-making have a significant positive correlation with CsADH gene expression [21]. The green-making process includes turnover and spreading. Some studies have found that most of the aroma compounds are formed during static spreading after turnover, and transcription factors play an important role in the response to mechanical injury during the manufacturing process of RGT [22].

During the withering and turning over treatments of oolong tea, the expression and abundance of phenylalanine ammoniase, peroxidase and polyphenol oxidase were positively correlated with the dynamic changes in their corresponding catalytic products, such as carbohydrates, amino acids and flavonoids [23]. Sun withering can induce the expression of protease in phenylpropanoid biosynthesis, flavonoid biosynthesis, phenylalanine metabolism, sesquiterpenoid and triterpenoid biosynthesis and other metabolic pathways in tea; reduce the content of flavonoids, xanthine alkaloids and other metabolites; accumulate volatile terpenoids and phenylpropanoids/benzenoids; and promote the formation of good flavor characteristics in oolong tea [24]. Spraying exogenous methyl jasmonate on fresh tea leaves could significantly affect the expression of the α-linolenic acid degradation pathway, terpene backbone biosynthesis and phenylalanine metabolism-related proteins and promote the synthesis of green leaf volatiles, volatile terpenoids and volatile phenylpropanoids/benzenoids [25].

The combined transcriptome and metabolome analysis method was used to investigate the changes in volatile components, the expression of related genes in the anabolic pathway of the main aroma compounds and the function of single key genes during the processing of RGT. However, transcriptional regulation is a very complex biological process, and there are few reports on the changes in post-transcriptional protein regulation during the RGT manufacturing process. The aim of this research was to use transcriptome, proteome and gas chromatography–time-of-flight mass spectrometry (GC-TOF-MS) to explore the mechanism of aroma formation during the green-making process of RGT, and to analyze the differentially expressed genes (DEGs) and proteins (DEPs) and the protein interaction network, which can provide an exhaustive and precise understanding of the flavor formation process and further elucidate the influence of the green-making process on the characteristic quality metabolites of RGT.

## 2. Materials and Methods

### 2.1. Sample Preparation and Collection

The experimental tea plantation of this study was the Jiulongshan base in the Wuyi Mountain area of Fujian Province (117.95° E, 27.71° N), at an altitude of 260 m. First, 20 kg fresh leaves (cultivar Rougui) were collected, and the standard of picking was three leaves and one bud. Then, they were processed according to the traditional production method of RGT. Briefly, the freshly plucked tea leaf samples were exposed to sun withering for 30 min. Subsequently, they were averagely divided into two groups. One group was subjected to manual green-making, including seven shaking steps, and the number of turns during shaking was 10, 20, 40, 60, 80, 120 and 300, respectively. The interval between two successive shaking steps was 1 h, and the green-making process lasted about 8 h. At the same time, under the same environmental conditions, another group of fresh leaves was spread for 8 h without shaking; the temperature during green-making was 23–25 °C and the relative humidity was 60–70%. Subsequently, the tea leaves of the two groups were produced into finished tea after fixation, rolling and drying. The samples included sun-withered leaves (SY); the experimental group (LY group) (1st shaking (LY1), 3rd shaking (LY3), 5th shaking (LY5), 7th shaking (LY7)); the control group (CK group) (1st spreading (CK1), 3rd spreading (CK3), 5th spreading (CK5), 7th spreading (CK7)); the finished tea with green-making (ZM); and the finished tea without green-making (DM). Three biological replicates for each sample point were obtained, immediately frozen and fixed in liquid nitrogen, and then stored in a −80 °C refrigerator for study. Samples SY, LY7, CK7, ZM and DM were taken to detect and analyze the volatile compounds. Samples SY, LY1, LY3, LY5, LY7, CK1, CK3, CK5 and CK7 were taken for transcriptome analysis. Samples SY, LY7 and CK7 were taken for proteomic analysis.

### 2.2. Analysis of Aroma Components

After the samples were fully ground, 2 g of each sample was weighed and placed in a headspace vial. The aroma substances were extracted by solid-phase microextraction (SPME), and the extracted aroma substances were detected by GC-TOF-MS (Agilent 7890B/Pegasus HT TOF MS, Agilent, Palo Alto, CA, USA/LECO, St. Joseph, MI, USA). In brief, firstly, 2 g of powdered tea was placed into an 18 mL headspace vial. Additionally, 10 μL 2-octanol (10 mg/L) was added as an internal standard. Subsequently, 5 mL of boiling deionized water was added into the vial to brew the sample. The vial was then sealed with a silica gel diaphragm and allowed to incubate in a water bath at 80 °C for 30 min. Following this, the SPME fibers (2 cm, 50/30 μm, PDMS/DVB, Stableflex, Sigma-Aldrich, Shanghai, China), which had been activated in advance, were then inserted into the headspace of the vial and kept for 60 min to ensure the complete extraction of the volatiles [26,27]. Following extraction, the fiber was manually inserted into the gas chromatograph sample inlet, which was set to 250 °C, to desorb the volatiles. Specifically, an Agilent 7890B gas chromatograph (Agilent, Palo Alto, CA, USA) equipped with a Rxi-5si1MS column (Agilent, Palo Alto, CA, USA) was used. The GC injection method applied here was no-split injection, where the temperature of the injection port was set to 250 °C. The carrier gas was high-purity helium (purity > 99.999%) with a flow rate of 1 mL/min. Initially, the temperature was maintained at 50 °C for 5 min, followed by a gradual temperature increase to 210 °C (at a rate of 3 °C/min) and this temperature was maintained for 3 min. Subsequently, the temperature was raised further to 230 °C (at a rate of 15 °C/min). For MS detection, the EI ion source temperature was set to 250 °C, the scan rate was 10 spectra/s, the solvent delay time was 5 min, the EM voltage was 70 eV, and the MS scan range was 45 to 600 amu. The experiment was repeated three times. At the same time, the same amount of each sample was mixed to prepare QC. During the data acquisition process, one QC sample was collected for every three samples to test the stability of the instrument. The GC-TOF-MS data were integrated with the Chroma TOF version 4.51.6 (American LECO Company, St. Joseph, MI, USA) software. The volatile peaks were identified by matching the National Institute of Standards and Technology (NIST) mass spectral database and the retention index (RI, determined by n-alkanes C7–C40).

### 2.3. Transcriptomic Analysis on RGT Samples

The samples were ground in liquid nitrogen and were used to extract the total RNA in three replicates of each sample. The RNA extraction method referred to the polysaccharide polyphenol plant total RNA extraction kit (Tiangen Biology Co., Ltd., Beijing, China). The integrity of the extracted RNA was analyzed by agarose gel electrophoresis; the RNA concentration was accurately quantified by a Nanodrop 2000 (Thermo Fisher Scientific, Waltham, MA, USA) microspectrophotometer; and the total RNA quality was accurately detected by an Agilent 2100 (Agilent, Palo Alto, CA, USA).

After extracting the total RNA, the eukaryotic mRNA was enriched using Oligo (dT) beads, while the prokaryotic mRNA was enriched by depleting the rRNA with the Ribo-Zero TM Magnetic Kit (Epicentre, Madison, WI, USA). Subsequently, the enriched mRNA underwent fragmentation into short fragments using a fragmentation buffer and was reverse-transcribed into cDNA with random primers. The second-strand cDNA was then synthesized using DNA polymerase I, RNase H, dNTP and a buffer. Following this, the cDNA fragments were purified using the QiaQuick PCR extraction kit (Beckman Coulter, Beverly, MA, USA) and underwent end repair, the addition of poly(A) and ligation to Illumina sequencing adapters. The resulting ligation products were selected based on size using agarose gel electrophoresis, amplified through PCR and sequenced using the Illumina HiSeq TM 2500 by Gene Denovo Biotechnology Co., Ltd. (Guangzhou, China).

Gene abundance was quantified by the RSEM software v1.3.3. The gene expression level was normalized by using the fragments per kilobase of transcript per million mapped reads (FPKM) method. Principal component analysis (PCA) was performed with the R package gmodels (http://www.r-project.org/ accessed on 29 October 2022). During this process, we identified genes with a fold change ≥ 2 and a false discovery rate (FDR) < 0.05 in a comparison as significant DEGs. The DEGs were then subjected to an enrichment analysis of the Gene Ontology (GO) functions and Kyoto Encyclopedia of Genes and Genomes (KEGG) pathways.

### 2.4. Proteome Analysis of Tea Samples

Three independent biological replicates were performed for samples SY, LY7 and CK7. Proteins were extracted using the cold acetone method. Samples were ground to powder in liquid nitrogen and then dissolved in 2 mL lysis buffer (8 M urea, 2% SDS, 1× protease inhibitor cocktail) and sonicated on ice for 30 min. After centrifugation at 13,000× *g* for 30 min at 4 °C, the supernatant was transferred to a fresh tube. Proteins were precipitated with ice-cold acetone at −20 °C overnight and cleaned three times before being re-dissolved in 8 M urea by sonication on ice. The protein quality was examined with SDS-PAGE.

The BCA Protein Assay Kit was used to determine the protein concentration of the supernatant. Then, 100 μg protein per condition was adjusted to a final volume of 100 μL with 8 M urea and incubated with DTT at 37 °C for 1 h. Then, the samples were centrifugated at 14,000× *g* for 10 min in a 10 K ultrafiltration tube (Millipore, Burlington, MA, USA). Next, 120 μL of 55 mM iodoacetamide was added to the sample and incubated for 20 min, while protected from light, at room temperature. For each sample, 8 M urea was replaced with 10 mM triethylammonium bicarbonate (TEAB) by centrifugating the mixture three times. Proteins were then subjected to tryptic digestion with sequence-grade modified trypsin (Promega, Madison, WI, USA) at 37 °C overnight. Then, the digested samples were centrifugated at 13,500× *g* for 12 min, dried in a vacuum and dissolved in 500 mM TEAB. The resultant peptide mixture was labeled with TMT tags for 2 h at room temperature. The labeled samples were combined and dried in a vacuum.

Tandem mass spectra was extracted and the charge state deconvoluted and deisotoped by Mascot Distiller version 2.6. Then, the mass spectrometry data were transformed into MGF files with Proteome Discovery 1.2 (Thermo, Pittsburgh, PA, USA) and analyzed using the Mascot search engine (Matrix Science, London, UK; version 2.3.2). Protein identifications were accepted if they could achieve an FDR less than 1.0% via the Scaffold Local FDR algorithm. Protein quantification was carried out in those proteins identified in all samples with unique spectra ≥ 2. Proteins with a fold change in a comparison > 1.2 or < 0.83 and an unadjusted significance level *p* < 0.05 were considered differentially expressed.

### 2.5. Quantitative Real-Time PCR (qRT-PCR)

The polysaccharide polyphenol total RNA extraction kit (Tiangen Biology Co., Ltd., Beijing, China) was used to extract the total RNA, and the Fastking gDNA Dis-PelllingRT SuperMix kit (Tiangen Biology Co., Ltd., Beijing, China) was used to synthesize cDNA as a real-time fluorescent quantitative PCR template, using three biological replicates. Using CsGAPDH as the internal reference gene, the Applied Biosystems fluorescence quantitative PCR instrument was used to perform qRT-PCR. The cDNA diluted 101, 102, 103, 104 and 105 times was used as the template for qPCR. The prepared qPCR reaction solution is shown in Appendix A, and the total reaction system totaled 20 μL. The 2^−∆∆Ct^ method was used to calculate the gene expression levels. The SnapGene 6.1 software was utilized to generate the gene-specific primers (Appendix A).

### 2.6. Statistical Analysis

The Origin 2022b software was used to calculate the mean value and variance of the data and perform the principal component analysis (PCA). The Venn diagrams, volcano plots, heat maps, KEGG metabolism pathway enrichment diagrams and correlation analysis were obtained with the R software v4.2.3. Protein–protein interaction networks (PPI) were evaluated using the Search Tool for the Retrieval of Interacting Genes (STRING) database.

## 3. Results

### 3.1. Analysis of Content of Volatile Compounds in Tea Leaves

In this study, the aroma compounds of SY, LY7, CK7 and the finished tea were as shown in Appendix A, including 15 esters, 12 alcohols, 5 aldehydes, 3 ketones, 4 olefins and 3 heterocycles. Compared with SY, esters, alcohols, aldehydes and ketones with pleasant flowery and fruity aroma compounds in LY7 and CK7 were significantly increased, and the content of aroma compounds in LY7 was significantly higher than that in CK7. Nerolido, jasmine lactone, indole, 2-isopropenyl-5-methyl-4-hexenol, farnesene, methyl salicylate, jasmonone, acetic acid hexyl ester, (E)-3-hexenyl butyrate and hexanoic acid hexyl ester were the main aroma compounds that had accumulated at the end of green-making and spreading, all of which had a strong floral and fruity aroma.

Terpene derivatives are an important component of the tea aroma and one of the main substances for the formation of RGT’s fruity aroma characteristics. Volatile terpenoids were detected in the samples, including nerolido, farnesene, linalool and oxidation products, β-ocimene and so on. The content of nerolido in LY7 and in CK7 showed a significant difference. Farnesene is a common sesquiterpene aromatic substance in RGT [28]. The present study detected that α-farnesene and (E)-β-farnesene were increased at the end of green-making and spreading. Linalool is an important monoterpene aroma substance in tea [29]. Two linalool oxides and dehydrogenated linalool were detected in this experiment, and, compared with SY, the content of these volatile substances in LY7 and CK7 was increased.

The phenylalanine metabolic pathway is an important pathway for the synthesis of aroma compounds in plants, and the related aroma compounds are phenylethyl alcohol, benzyl alcohol, benzaldehyde, phenylacetaldehyde, methyl salicylate and so on. After the RGT fresh leaves were withered, the aroma compounds were significantly increased during the green-making and spreading process, and their content in LY7 was significantly higher than that in CK7. Phenylalanine produces trans-cinnamaldehyde under the action of phenylalanine ammonia lyase, and methyl salicylate is synthesized by a multi-step reaction [30]. Compared with SY, the methyl salicylate content in LY7 and CK7 increased significantly, which indicated that mechanical injury and dehydration could promote the synthesis of related products in the phenylalanine metabolic pathway. Studies have shown that mechanical injury and low-temperature stress can promote the accumulation of indole, jasmine lactone, nerolido and volatile fatty acid derivatives during the RGT manufacturing process [31]. In this study, jasmine lactone, jasmone, aliphatic alcohols and esters, which were related to the oxidative degradation of fatty acids [32], showed an increasing trend throughout the manufacturing process, and this was most obvious during the green-making process. Continuous damage in the process of green-making induces the transcription of genes related to jasmonic acid biosynthesis, which can promote the formation of terpenoids [33]. In this study, the content of jasmonic acid derivatives and the main volatile terpenoids in the LY group increased, which was beneficial for the formation of the quality characteristics of the floral and fruity aroma.

The results of GC-TOF-MS showed that long-term mechanical injury and dehydration could significantly increase the terpene terpenoids and ester compounds, which are the main aroma components of RGT [34], and this was more obvious during the green-making process. While the content of most aroma components decreased in DM and ZM, this may have led to the loss of the volatile compounds under the action of the subsequent high temperatures. In this study, the sensory evaluation of the finished tea revealed that ZM had a more fragrant and lasting aroma and an obvious flowery and fruity flavor, while DM showed a clean and refreshing aroma, which was directly related to the type and content of the aroma compounds. In addition, the most prevalent aroma substances in RGT mainly include nerolidol, jasmine lactone, aliphatic alcohols, indole and so on, among which the ester components are rich, which make its fruity aroma characteristics more prominent. Meanwhile, the main aromatic substances of Tieguanyin oolong tea are linalool, nonanal, indole, nerolidol, farnesene and phenethyl isovalerate, which combine to its floral and green grassy aroma [35].

### 3.2. Analysis of Transcriptomic Profiles of Tea Samples

A total of 33,932 genes were obtained via the transcriptome sequencing of all samples, including 30,458 known genes, accounting for 89.76%, and 5076 unknown new genes. The results of the PCA showed (Figure 1A) that two principal components could distinguish all samples, and the variance of the two principal components was 73.6% and 11%, respectively, collectively accounting for 84.6% of the total variability of the volatile compounds, which was sufficient to interpret the information of the volatile components in each tea sample. Figure 1B reveals that the distribution patterns of different samples were similar, in which the distribution density of the LY group was lower than that of the CK group. The transcriptional data further showed the number of significantly different genes in CK at different stages and SY gradually increased, and the number of downregulated genes was greater (Figure 1C,D) (*p* < 0.05). Compared with SY, LY7 had the largest number of significant differences (3849 genes were upregulated and 9299 genes were downregulated in expression). Then, the expression of most genes in the fresh tea leaves decreased after long-term mechanical injury and dehydration, and the effect of the green-making process on the gene expression in the fresh tea leaves was more obvious. These results were similar to the previously obtained transcriptome data of RGT that was treated with different degrees of green-making [36].

The Venn diagram (Figure 1E) showed that LY5 vs. CK5 and LY7 vs. CK7 had the largest number of DEGs (3923 genes), and the four comparison groups shared 263 DEGs. The KEGG pathway enrichment results (Figure 1F) showed that there were significant differences in glucosinolate biosynthesis, phenylpropanoid biosynthesis and glutathione metabolism. The metabolic pathways related to aroma formation are enriched by DEGs, including monoterpenoid biosynthesis, alpha-linolenic acid metabolism and terpenoid backbone biosynthesis. These metabolic pathways are important in the formation of RGT’s aroma and should be the focus of further study.

### 3.3. Proteome Profiling of Proteins Differentially Expressed in Response to Green-Making Process

SY, LY7 and CK7 were extracted and qualitatively quantified by the TMT labeling technique to explore the mechanism of influence of the green-making process on RGT’s aroma metabolism from the perspective of proteomics. A total of 623,142 secondary spectra were identified in all samples, among which 36,759 peptide spectra and 29,159 unique peptide spectra were identified. A total of 19,376 peptides were identified, including 16,650 unique peptides and 5242 proteins. The relative molecular weight distribution statistics of the proteins are shown in Figure 2A; 49.93% of the proteins were 0–50 kDa, 39.97% were 51–100 kDa and 10.10% were greater than 100 kDa. Most of the peptide lengths in the proteins were distributed among 6–13 amino acid residues, accounting for 73.41% of the total peptide number (Figure 2B). The proportions of proteins with peptide sequence coverage below 10%, 10–20%, 20–30%, 30–40% and 40–100% were 50.32%, 30.14%, 12.57%, 4.63% and 2.34%, respectively (Figure 2C). The proportion of proteins with peptide sequence coverage below 20% accounted for 80.46% of the total proteins. The number distribution of the peptides contained in the proteins identified is shown in Figure 2D. The number of proteins containing one to five peptide segments was 3377, accounting for 70.49%; the number of peptides containing 6 to 10 peptide segments was 1020, accounting for 21.29%; and the number of peptides containing 11 or more peptide segments was 394, accounting for 8.22%, indicating that the number of peptides contained in most proteins was within five, followed by 6–10.

In this study, a total of 5242 proteins were identified, which were screened according to the conditions of a more than 1.2 times difference and a *p*-value less than 0.05. The results showed that there were 635 DEPs between SY and CK7, and the number of upregulated and downregulated proteins was 378 and 257, respectively. There were 729 DEPs between SY and LY7, and the number of upregulated and downregulated proteins was 495 and 234, respectively. There were 385 DEPs between CK7 and LY7, and the number of upregulated and downregulated proteins was divided into 270 and 115 (Figure 3A–D), which was consistent with the metabolic data. Long-term green-making can significantly change the metabolic activity of substances in fresh tea leaves, improve the intensity of protein synthesis and promote the production of metabolic substances.

In living organisms, proteins usually do not perform their functions independently, but many different proteins cooperate with each other to complete a biological function or participate in a metabolic pathway. The significant enrichment of pathways can identify the major biochemical metabolism and signal transduction pathways involved in DEPs. The KEGG enrichment analysis (Figure 3E,F) showed that the differential proteins between SY, LY7 and CK7 were involved in metabolic pathways including flavonoid biosynthesis, alpha-linolenic acid metabolism, linoleic acid metabolism and fatty acid metabolism, the biosynthesis of unsaturated fatty acids, glyoxylate and dicarboxylate metabolism, carotenoid biosynthesis, phenylalanine metabolism and terpenoid backbone biosynthesis. Compared with CK7, DEPs in LY7 involved more significant aroma metabolic pathways and synthesized more abundant aroma compounds. The biological metabolism of fatty acids can produce a large number of aroma precursors, which participate in the metabolic pathways of linoleic acid and α-linolenic acid, and these unsaturated fatty acids are oxidized and degraded to form rich aliphatic aroma compounds; one of the branch pathways of α-linolenic acid is related to the formation of jasmonic acid, and the accumulation of jasmonic acid promotes the expression of terpenoid synthesis genes and the synthesis of terpenes [37]. At the same time, the metabolism of unsaturated fatty acids, terpenoid backbone biosynthesis and phenylpropanoid biosynthesis can provide substrates for the synthesis of volatile esters, thus forming the flowery and fruity flavor of oolong tea.

### 3.4. Correlation Analysis of Proteome and Transcriptome

The correlation analysis of the transcriptome and proteome is used to evaluate the post-transcriptional regulation or translational regulation of genes in organisms and the metabolic process [38]. The proteins identified were compared with the transcription data of the corresponding gene, and a nine-quadrant diagram was used to show the changes in the genes and proteins. The horizontal coordinate was the protein difference multiple (log2 value), and the vertical coordinate was the difference multiple of the transcriptome (log2 value); each point represented a gene or protein, and a black dot represented the proteins and genes without a difference. Blue dots indicated differential expression in the proteome but no differential expression in the transcriptome, green dots indicated differential expression in the transcriptome but no differential expression in the proteome, and red dots indicated both differential genes and differential proteins. The graph was divided into nine quadrants by dashed lines on the horizontal and vertical coordinates. The dashed lines on the horizontal coordinates represented the difference multiple thresholds of the transcriptomes, while the dashed lines on the vertical coordinates represented the difference multiple thresholds of the proteome. Genes or proteins with significant differences were represented outside the threshold lines, while genes or proteins with no significant differences were represented inside the threshold lines. The genes and proteins in the third and seventh quadrants showed the same differential expression pattern, indicating that the level of post-transcriptional translation was not regulated or less regulated. The first, second and fourth quadrants indicated that the abundance of protein expression was lower than that of transcription, indicating that transcriptional regulation was regulated by the translation level, and transcriptional regulation target genes led to the inhibition of protein translation. The sixth, eighth and ninth quadrants indicated that the abundance of protein expression was higher than that of transcription, demonstrating that transcription was regulated or influenced by the level of translation. The correlation analysis results of each comparison group are shown in Figure 4A–C, and the consistency of proteome and transcriptome expression, respectively, was low. The number of proteins and genes in SY and CK7, SY and LY7 and CK7 and LY7 was 200, 221 and 53, respectively. The number of proteins and genes with lower protein expression abundance than the transcriptome among the three comparison groups was 317, 295 and 145, respectively. There were 913, 1316 and 810 proteins and genes with high expression abundance among the three comparison groups, respectively. There were 1665, 1263 and 2087 proteins with no significant differences between the three comparison groups. In conclusion, there were significant differences in the expression patterns of proteins and genes in RGT after a long period of green-making, and the regulation of post-transcriptional translation plays an important role in the metabolic activity of organisms.

An enrichment analysis of the metabolic pathways of differential genes or proteins in SY and LY7 in the comparison groups was performed (Figure 4D–F). The results showed that the pathways enriched in quadrant 3 were phenylpropanoid biosynthesis, the biosynthesis of secondary metabolites, zeatin biosynthesis and so on. The enriched pathways in quadrant 6 included ribosome, endocytosis and lysine biosynthesis and so on, and the enriched pathways in quadrant 9 included peroxisome, ribosome and photosynthesis antenna proteins and so on. Alpha-linolenic acid metabolism is an important pathway related to aroma metabolism and was co-enriched in these three quadrants, suggesting that the green-making process can promote the accumulation of proteins related to this metabolic pathway. The main reason is that abiotic stresses such as mechanical injury and dehydration activate the genes/proteins related to the synthesis of jasmonic acid in alpha-linolenic acid metabolism, thus inducing the accumulation of jasmonic acid, which is one of the most rapidly emerging stress signals in plants [33].

### 3.5. Analysis of DEGs and DEPs Related to Biosynthesis of Aroma Components

In addition to the changes in the main aroma components of the samples and the KEGG pathways enriched by differential genes and proteins, we further investigated the expression patterns of key genes and proteins related to the aroma synthesis of RGT (α-linolenic acid metabolism, terpenoid biosynthesis and the phenylalanine metabolism pathway). The expression of the differential genes and proteins related to the α-linolenic acid metabolism pathway detected in this research between samples is shown in Figure 5A. The upregulation of the lipoxygenase (LOX) gene was more obvious in the CK group. In the process of green-making, the LOX gene showed a trend of first increasing and then decreasing and reached the lowest level at the end of green-making, which was essentially consistent with previous studies [39]. Two coding proteins (TEA011765.1 and TEA012289.1) were detected in the proteome data. LOX was upregulated about one to two times at the end of green-making, and the upregulated range was higher in the CK group. Some studies have shown that mechanical injury, tissue injury, drought and other stress conditions can improve the transcription and expression of the enzyme genes and improve the enzymes’ activity [40,41,42]. LOX is a key rate-limiting enzyme in the alpha-linolenic acid metabolic pathway, and this enzyme gene is induced under mechanical injury and dehydration and can promote the synthesis of related metabolites [43,44]. Hydroperoxide lyase (HPL) and alcohol dehydrogenase (ADH) catalyze the formation of green leaf volatiles (GLVs). Studies have shown that when plants are subjected to abiotic stresses such as mechanical injury, the expression levels of the HPL gene and protein are increased, and the expression levels of the HPL gene and protein are improved through ADH and alcohol acyltransferases (AAT), which are further transformed into corresponding alcohols, esters and other volatiles to produce defense reactions [45,46]. The HPL gene’s expression was significantly increased in the CK group (*p* < 0.05), while the ADH gene’s expression was generally higher at the end of green-making, in which TEA029320.1 was upregulated to the maximum value, while no relevant protein data were detected by proteomic analysis. Another branch of α-linolenic acid metabolism, the allene oxide synthase (AOS) pathway, is involved in the synthesis of jasmonic acid and its derivatives, and multiple genes encoding the same enzyme were detected in this branch of the pathway, but their expression was not completely consistent in all samples. The expression of the AOS protein was significantly upregulated at the end of green-making and spreading, while it was downregulated in the transcriptome data (*p* < 0.05). The allene oxide cyclase (AOC) gene increased at first and then decreased in the process of green-making, and most of the coding genes decreased after the seventh shaking step, yielding slightly higher levels than in the CK group. Six 12-oxophytodienoic acid reductase (OPR) genes were detected, among which TEA025907.1, TEA026806.1 and TEA029800.1 were highly expressed in LY compared to CK. One protein’s expression trend was consistent with the transcriptome data, while the other protein was inconsistent. It was found that an OPCL gene was highly expressed in SY and was the highest at the end of green-making, but the expression trend of its coding protein showed the opposite pattern. Acyl-CoA oxidase (ACX) is the rate-limiting enzyme of fatty acid β-oxidation, and its expression increased during the manufacturing process. Two proteins (TEA017099.1 and TEA019216.1) were expressed at the highest levels at the end of green-making. The expression of the enoyl-CoA hydratase (MFP2) and acetyl-CoA acyltransferase (ACAA) genes was significantly upregulated under continuous mechanical injury and dehydration, while the expression trends of the two coding proteins were the opposite to those in the transcriptional group (*p* < 0.05). Some studies have shown that shaking and rocking can upregulate the LOX gene to increase the enzyme activity, thus promoting the biosynthesis of jasmine lactone, leading to the accumulation of volatile fatty acids [19]. In this study, the content of jasmine lactone in green-making also increased significantly (*p* < 0.05). To summarize, the green-making and spreading process had significant effects on the α-linolenic acid metabolic pathway, most of the related genes were upregulated, and the expression abundance of some proteins (AOS, OPCL, ACX) was higher than that in the transcriptional group, which was affected by the translation level’s regulation or accumulation after transcription. This was beneficial to promote the synthesis of volatile aliphatic and jasmonic acid derivatives.

Terpenoid aromatic compounds are the main substances that form the floral and fruity characteristics of RGT, among which monoterpene and sesquiterpene have the greatest influence. Most of the differential genes in this experiment were concentrated in the terpenoid backbone biosynthesis pathway (Figure 5B). This pathway can be divided into the mevalonate pathway (MVA) pathway and methylerythritol 4-phosphate pathway (MEP) pathway, which are carried out in the cytoplasm and plastid, respectively [47]. In the cytoplasm, hydroxymethylglutaryl-CoA synthase (HMGS) is the rate-limiting enzyme of the MVA pathway [48]. Two genes regulating HMGS were extracted from the samples and upregulated in the process of green-making and spreading. The upregulation effect of green-making was obvious, and the trend of protein expression was the same. At the same time, 3-hydroxy-3-methyl glutaryl coenzyme A reductase (HMGR), the mevalonate kinase gene (MVK), the phosphomevalonate kinase gene (PMVK) and the methylglutarate diphosphate decarboxylase gene (MVD) in the MVA pathway were all upregulated during green-making, and they reached the highest levels at the end of green-making. The upregulation of related differential genes in LY was higher than that in CK. In the plastid, deoxyxylose-5-phosphate synthase (DXS) is a key rate-limiting enzyme in the regulation of monoterpene accumulation in the MEP pathway, and its genes are upregulated in Euphorbia angustifolia. In addition, the 1-deoxy-D-xylulose 5-phosphate reductoisomerase (DXR), 2-C-methyl-D-erythritol-2,4-cyclodiphosphate synthase (ISPF), (E)-4-hydroxy-3-methylbut-2-enyl-diphosphate synthase (ISPG) and (E)-4-hydroxy-3-methylbut-2-enyl pyrophosphate reductase (ISPH) genes were upregulated under the mechanical injury and dehydration of green-making and spreading, and the upregulation was more significant in the CK group. ISPG and ISPH are key regulatory genes for the synthesis of monoterpene and sesquiterpenes’ common precursors, dimethylallyl diphosphate (DMAPP) and isopentenyl diphosphate (IPP) [49]. It was found that the expression trend of the ISPG protein (TEA013763.1) was consistent with the transcriptome data. The expression of the 2-C-methyl-D-erythritol-4-phosphate cytidyltransferase (ISPD) gene decreased gradually with green-making and spreading, while the expression of the 4-(cytidine-5′-diphosphate)-2-c-methyl-d-erythritol kinase (ISPE) gene increased significantly at the first stage of green-making and spreading (*p* < 0.05) and then decreased gradually. However, the expression of the isopentene diphosphate isomerase gene (IDI), which catalyzed the transformation of DMAPP and IPP, was upregulated, and the expression of two genes (TEA018159.1 and TEA030524.1) increased at the end of green-making. Six geranylgeranyl diphosphate synthase (GGPS) genes showed no obvious upregulation in the process of green-making; only one corresponding protein (TEA003988.1) was upregulated, and three genes encoding GGPS (TEA006109.1, TEA018068.1 and TEA019181.1) were significantly upregulated at the end of spreading (*p* < 0.05). Geranyl-pp, synthesized by GGPS, is a precursor of monoterpenes. Under long-term mechanical injury and dehydration, the alcohol expression of the key enzyme genes in the terpenoid skeleton synthesis pathway maintained the trend of upregulation during the green-making and spreading process, and the expression level of cytoplasmic MVA pathway-related genes was more obvious than that of the plastid MEP pathway in the green-making process.

In the monoterpene and sesquiterpene biosynthetic pathway, the expression intensity of the sesquiterpene synthetase gene was higher than that of the monoterpene synthetase gene in the green-making process, and the expression intensity of the sesquiterpene synthase gene reached the highest level at the end of the fifth shaking step. Most of the monoterpene synthase genes reached the highest levels at the end of the first shaking step and then decreased. Most of the genes involved in sesquiterpene synthesis were upregulated after tea picking, while most of the genes involved in monoterpene synthesis were downregulated [50]. Our results are similar to those of previous studies. Studies have shown that multiple mechanical injuries during the processing of RGT can promote an increase in neroli tertiary alcohol synthase gene expression and the accumulation of a large amount of neroli tertiary alcohols, and the synergistic effect of low temperatures and mechanical injury is more conducive to the accumulation of neroli tertiary alcohols [51]. In conclusion, the green-making process of RGT was more conducive to the expression of sesquiterpene aroma synthesis-related genes, while the CK group promoted the expression of monoterpene aroma synthesis-related genes. At the same time, the data on the content of volatile compounds showed that the content of linalool in the LY group was lower than that in the CK group, while the content of nerolidol was higher than that in the CK group, which was consistent with the gene expression patterns.

Phenylalanine is a precursor of tea aromatic compounds, and its metabolic pathway is regulated by several genes. The differential genes that are enriched in the metabolic pathway are mainly related to the synthesis of phenylpyruvate. Phenylalanine is induced by aspartate aminotransferase (ASP) and tyrosine aminotransferase (TAT) to form phenylpyruvate, which is catalyzed to generate phenylacetaldehyde, phenylethyl alcohol and so on [52]. In addition, phenethylamine can also produce phenylacetaldehyde under the catalysis of primary amine oxidase (tynA) (TEA029368.1). After sun withering, the ASP gene expression gradually increased under the effects of continuous mechanical injury and dehydration, and it reached the highest level at the end of green-making and spreading. The green-making process had a more obvious effect on the upregulation of the gene, and a coding protein (TEA024318.1) with a similar expression pattern was detected. In addition, a bifunctional aspartate aminotransferase and glutamate (PAT) gene and related proteins were significantly upregulated during the green-making process (*p* < 0.05), and the tynA gene (TEA029368.1) also showed a similar trend (Figure 5C). The green-making process promoted the synthesis of phenylpyruvate, which increased the aroma volatiles, compared to the spreading process. The content of phenylacetaldehyde and phenylethyl alcohol in the LY group in ZM and DM was higher than that in the CK group.

Five phenylalanine aminolyase genes (PAL) can catalyze phenylalanine to form trans-cinnamic acid, which gradually forms benzene ring compounds through other pathways and finally forms benzyl alcohol, benzoic acid and other aromatic compounds [30]. The PAL gene expression was mostly upregulated during green-making and spreading, but the expression trends of the three coding proteins were inconsistent with the transcripts. The content of benzyl alcohol and benzaldehyde in the LY group in ZM and DM was higher than that in the CK group. In addition, the expression levels of two tryptophan synthase (TSB) genes related to indole biosynthesis showed an increasing trend in the LY and CK groups, and the corresponding protein expression trend was consistent, but there was no significant difference between LY7 and CK7. During the processing of RGT, the TSB gene can be induced and upregulated, and some studies have shown that the indole content of its product can be increased only under the contrasting effects of the TSA and TSB genes [53], but no TSA gene was detected in this experiment.

### 3.6. Correlation Analysis of DEGs and DEPs Related to Fatty Acid Metabolism and α-Linolenic Acid Metabolism

The volatile compound data showed that, in addition to the large enrichment of nerolidol and indole, the aliphatic alcohols, esters, jasmone and jasmine lactone in the LY group were also significantly increased compared with the CK group (*p* < 0.05). Previous studies have shown that the fruity flavor of RGT is mainly based on six ester substances, such as (Z)-3-hexen-1-yl butyrate, (E)-3-hexen-1-yl butyrate, 5-hexenyl butyrate and so on. Due to abiotic stresses such as high temperatures and mechanical injury, the expression of structural genes in the LOX pathway of unsaturated fatty acids can be induced during the manufacturing process, thereby enhancing the activity of key enzymes and promoting the oxidative degradation of unsaturated fatty acids (linoleic acid and α-linolenic acid), further forming rich fatty aroma compounds and jasmonic acid compounds [54]. The degradation of fatty acids in RGT manufacturing and processing is conducive to the formation of fatty acid-derived volatiles (FADV) and produces cyclic aroma compounds such as methyl jasmonate, jasmonone and jasmonolide [55]. According to the transcriptome and proteome data, the expression of α-linolenic acid metabolism pathway-related genes was greater, and the expression abundance of differential proteins was higher. The fatty acid metabolic pathway is a key upstream metabolic process in the formation of FADV. The correlation analysis of the DEGs and DEPs in these two metabolic pathways further explained the metabolic patterns that promote the formation of the RGT aroma. The DEGs and DEPs related to fatty acid degradation, the biosynthesis of unsaturated fatty acids and α-linolenic acid metabolism were analyzed (Figure 6A), which showed that the expression levels of the PED1, LACS7, MFP, TESB and ACAA genes were significantly and positively correlated with most differential proteins, while the expression levels of the LACS3, LOX3.1, OPCL, FAD7 and JMT genes were significantly and negatively correlated with most differential proteins (*p* < 0.05). Differential proteins AOS, CAC3, fabD, ACX4, OPCL, LACS3 and OPR3 were closely related to most of the differential genes. The effect of fatty acid metabolism on the upstream metabolic precursors of volatile substances during the manufacturing process needs further study to explore the mechanism behind the floral and fruity aroma formation of RGT. During the green-making process of oolong tea, the fatty acid content decreased gradually, while the fatty acid-derived volatiles increased, and the fatty acid ester content reached the highest level at the fourth shaking step [55]. Appropriately increasing the number of turns in the green-making process can improve the synthesis of aroma esters and promote the formation of fruity aromas.

### 3.7. Protein–Protein Interaction Network Analysis

Proteins rarely act alone but form complexes or interact with other proteins to perform most biological functions [56,57]. The protein–protein interaction (PPI) is a network in which the proteins interact in a living organism, helping to uncover key core genes [58]. The interaction results for differential proteins related to fatty acid metabolism and α-linolenic acid metabolism are shown in Figure 6B, where they are divided into three groups. The first group contained 4 genes, the second group contained 12 genes and the third group contained 21 genes. ACX, MFP2, OPCL1 and PED1 in the first group were most associated with other interacting proteins and were the main nodes of the protein interaction network, while the interaction coordination mechanism and functions need to be further studied. The AOS branch of the α-linolenic acid pathway is related to the synthesis of jasmonic acid and its derivatives, and ACX, MFP2 and OPCL1 are related proteins regulating this branch pathway. In this metabolic pathway, ACX and MFP2 participate in β-oxidation, among which ACX is the rate-limiting enzyme. The β-oxidation of fatty acids is related to the synthesis of lactones, and the content of jasmine lactone and jasmone in RGT was significantly increased after green-making (*p* < 0.05), suggesting that ACX, MFP2 and OPCL1 are the key genes for the formation of volatile jasmonic acid derivatives. They were closely related to the co-expression of the second group of DEGs.

### 3.8. Quantitative Real-Time PCR (qRT-PCR) Validation

To verify the reliability of the RNA-seq transcriptome data, we randomly selected 12 genes participating in α-linolenic acid metabolism, the terpenoid biosynthesis pathway and the phenylalanine metabolism pathway for qRT-PCR analysis. The transcriptome results were consistent with the qRT-PCR analysis data (Appendix A). This indicated that the transcriptomic data were reliable. The results of qRT-PCR showed that the expression of most genes increased slightly after sun withering for a short time, and the expression of most genes in the LY group was higher than that in the CK group at the beginning of the third shaking step. The results showed that the synthesis of aroma substances induced by mechanical damage began to play a significant role in the middle stage of green-making, so it is necessary to control the technological parameters of green-making in order to promote the formation of a high-quality aroma in RGT.

## 4. Conclusions

In this study, the fresh leaves of RGT were used as experimental materials. After 8 h of green-making (including shaking and spreading) and 8 h of spreading (unshaken), respectively, the content of volatile components in the fresh leaves increased to different degrees, and the content of nerolidol, jasmine lactone, jasmone, phenylethyl-alcohol, benzyl alcohol, indole, hexyl hexanoate, (E)-3-hexenyl butyrate and 1-hexyl acetate in the LY group was significantly higher than that in the CK group. The sensory evaluation results showed that the aroma of ZM was stronger, and the flowery and fruity aroma was highlighted, while DM was clean and refreshing. The transcriptome data showed that the green-making and spreading process downregulated a large number of genes in the fresh leaves of RGT; the effect in the green-making process was more obvious. Meanwhile, the clustering results revealed that the gene expression pattern during the green-making and spreading process was similar. Compared with SY, the number of upregulated proteins at the end of green-making and spreading was greater than that of the downregulated proteins, and the number of differential proteins in the LY group was higher. Combined with the transcriptome and proteome data analysis, we found that long-term green-making and spreading could activate the upregulated expression of genes related to the aroma anabolism pathway, but the regulation of protein expression was not completely consistent. The mechanical injury in the process of green-making was more conducive to the positive regulation of the α-linolenic acid metabolism pathway, especially the branch pathway of AOS, followed by the MVA pathway of terpenoid backbone biosynthesis, which was essentially consistent with the change trend of the final metabolites. The fatty acid metabolic pathway occurs upstream of the α-linolenic acid metabolism pathway. The correlation analysis of the genes and proteins between the upstream and downstream metabolic pathways showed that there was a significant correlation between them. The analysis of the protein interaction network demonstrated that ACX, MFP2, OPCL1 and PED1 were the key proteins, and ACX was the rate-limiting enzyme of fatty acid β-oxidation. Previous studies have found that the tea plant ACX gene can be induced by mechanical damage, jasmonic acid treatment and herbivorous insects [59,60]. The manufacturing processes of oolong tea showed that tea made from fresh leaves that have been sucked by the small green leafhopper had a strong fruity and honey flavor. In this study, it is shown that ACX is a key protein, and we can highlight the effect of the regulatory mechanism of this gene in the production and manufacturing processes on the aroma quality of oolong tea.

## Figures and Tables

**Figure 1 foods-13-01315-f001:**
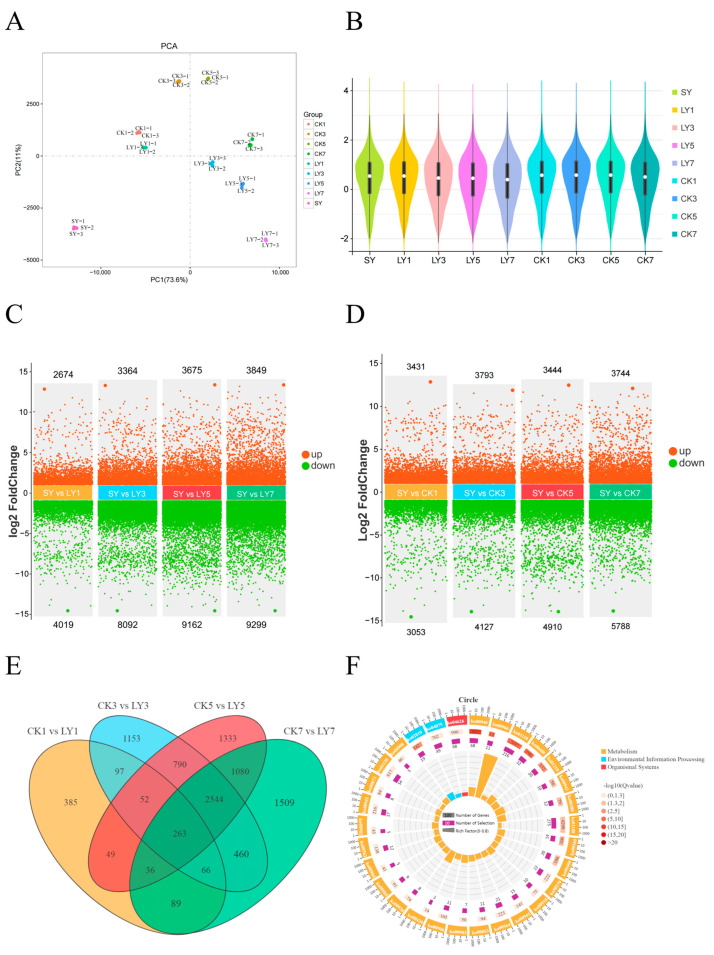
The profiles of the RNA-seq dataset from RGT samples during different processing stages. (**A**) The PCA score plot of different leaf samples based on the transcriptomic dataset. (**B**) The violin plot of different leaf samples. (**C**,**D**) The number of significant differential genes in various paired comparisons with the criteria of log2 |(fold change)| ≥ 1 and false discovery rate < 0.05. The numbers labeled on each comparison represent the number of up- or downregulated genes. (**E**) The Venn diagram of the genes quantified in the transcriptome. (**F**) The enrichment analysis of the differentially expressed genes between the comparison groups.

**Figure 2 foods-13-01315-f002:**
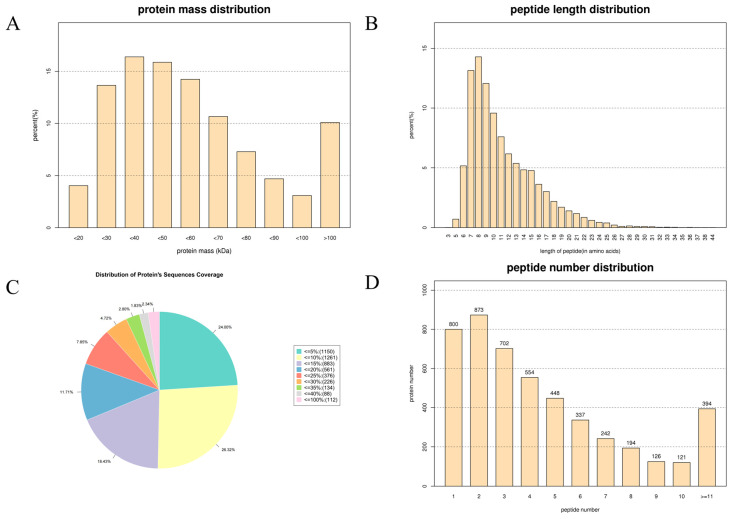
The profiles of the proteome dataset from RGT samples. (**A**) Statistics of protein mass distribution; (**B**) statistics of peptide length distribution; (**C**) distribution of protein sequence coverage; (**D**) statistics of peptide number distribution.

**Figure 3 foods-13-01315-f003:**
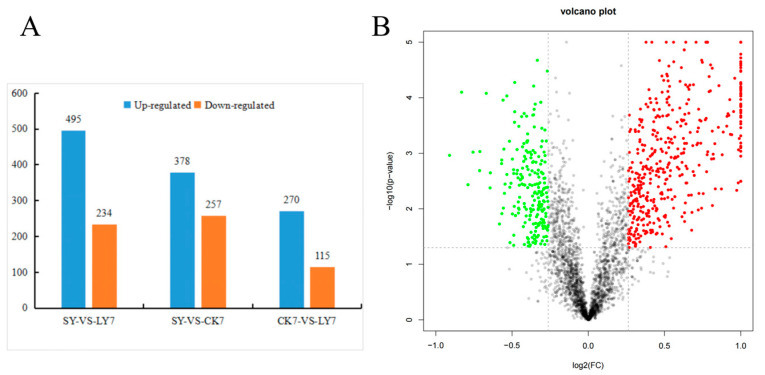
Overview of the differentially expressed proteins (DEPs) between SY, LY and CK. (**A**) Number of identified DEPs of different tea samples. (**B**–**D**) The volcanic map was drawn by using the fold change in DEPs between SY and LY7, SY and CY7, CY7 and LY7, respectively. (The *p*-value was obtained by a *t*-test and is used to show the significant differences between the two groups. The abscissa is the change fold (logarithmic transformation with base 2), the ordinate is a significant *p*-value (logarithmic transformation with base 10), red dots represent upregulated proteins and green dots represent downregulated proteins with significant differences (fold change was more than 1.2-fold and *p*-value < 0.05), and black dots represent proteins with no difference). (**E**,**F**) The significantly enriched KEGG pathways of the DEPs in SY and LY7, SY and CK7, respectively.

**Figure 4 foods-13-01315-f004:**
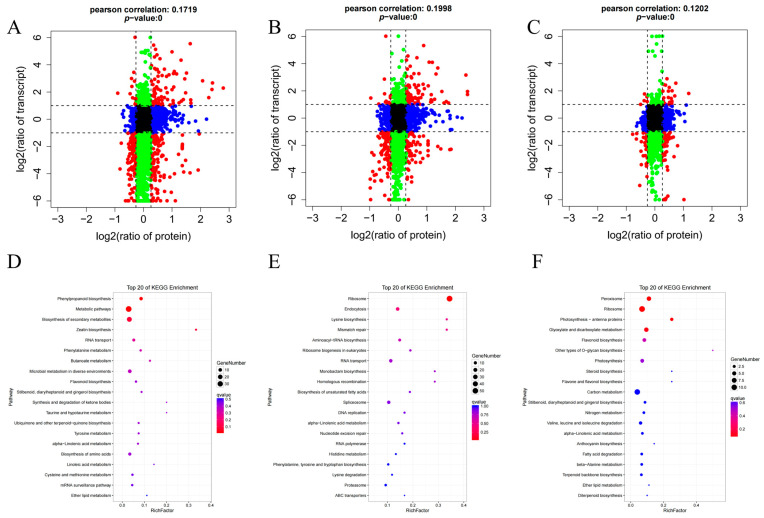
Combined transcriptome and proteome analysis between comparison groups. (**A**–**C**) A nine-quadrant map of the mRNA and protein associations between SY and LY7, SY and CY7, CY7 and LY7, respectively (each dot represents a gene or protein), red dots represent genes and proteins with significant differences (fold change was more than 1.2-fold), green dots represent genes with significant differences (fold change was more than 1.2-fold) and proteins with no difference, blue dots represent proteins with significant differences (fold change was more than 1.2-fold) and genes with no difference, black dots represent genes and proteins with no difference. (**D**–**F**) KEGG enrichment of DEGs and DEPs in quadrants 3, 6, 9, respectively.

**Figure 5 foods-13-01315-f005:**
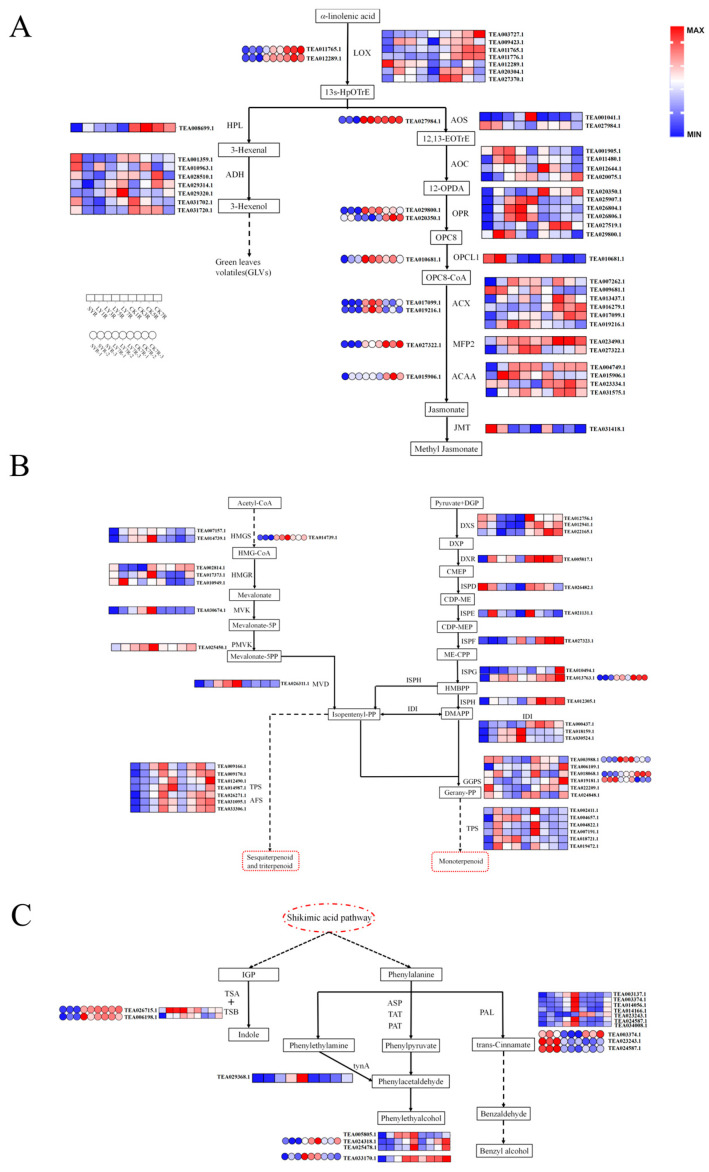
DEGs and DEPs involved in α-linolenic acid metabolism, terpenoid biosynthesis and phenylalanine metabolism pathway. (**A**) α-linolenic acid metabolism pathway; (**B**) terpenoid biosynthesis pathway; (**C**) phenylalanine metabolism pathway.Dashed boxes represent downstream metabolites and dashed circle represents upstream metabolic pathway.

**Figure 6 foods-13-01315-f006:**
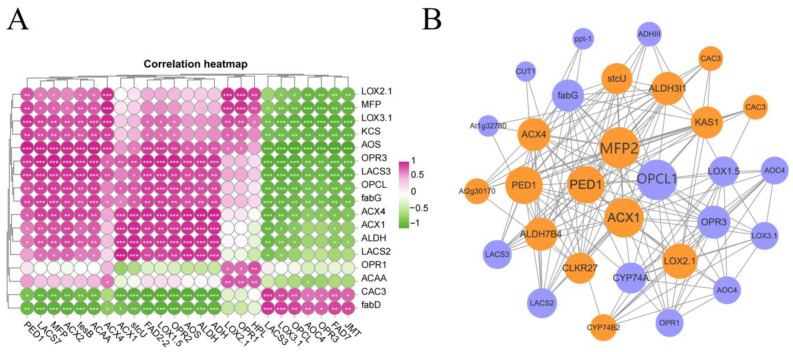
Correlation between DEGs and DEPs and qRT-PCR validate the reliability of the transcriptomic data. (**A**) Heat map of the correlations between the DEGs and DEPs involved in fatty acid metabolism and α-linolenic acid metabolism (the vertical coordinates are DEPs and the horizontal coordinates are DEGs. *: 0.01 < *p* < 0.05; **: 0.001 < *p* ≤ 0.01; ***: *p* ≤ 0.001). (**B**) Protein–protein interaction networks of DEPs (the orange nodes indicate consistent expression, and the blue nodes indicate inconsistent expression).

## Data Availability

The original contributions presented in the study are included in the article/Appendix A, further inquiries can be directed to the corresponding authors.

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
