# Peer review of "Effect of Mechanical Damage in Green-Making Process on Aroma of Rougui Tea"

_foods, 2024, doi:10.3390/foods13091315_

Round 1
Reviewer 1 Report
Comments and Suggestions for Authors
Dear Authors,
This study aimed to investigate the impact of the green-making process, including shaking and spreading, on the aroma profile of fresh leaves of RGT (a type of tea). By analyzing changes in volatile components and conducting sensory evaluations, the research sought to understand how different processing steps affect the concentration of key aroma compounds and the overall sensory qualities of the tea. The study combined transcriptome and proteome data to identify the genetic and protein expression changes associated with aroma biosynthesis, focusing on pathways like α-linolenic acid metabolism and terpenoid backbone biosynthesis. The goal of study was to elucidate the molecular mechanisms driving aroma formation in tea leaves during processing, providing insights for enhancing aroma characteristics through cultivation and processing techniques.
I believe that making the following revisions before publication will significantly improve the manuscript. Additional comments can be seen in the attached file. Also, the similarity index of the study is 32%, please reduce this rate.
introduction
This section falls short in discussing the impact of post-transcriptional protein regulation on the tea's aroma profile, indicating a gap in the current understanding of the aroma formation in RGT. Further research in this area could provide deeper insights into the intricate processes that contribute to the tea's distinct qualities.
Results and discussion
The section could benefit from a deeper exploration of the specific roles and mechanisms by which these compounds contribute to the overall aroma profile of the tea. While the analysis highlights the importance of terpenoids and the phenylalanine metabolic pathway in aroma formation, further clarification on how these compounds interact and influence each other to produce the final aroma would enhance the understanding of the complex aroma characteristics of Rougui Tea. Additionally, integrating comparisons with other tea varieties could provide a broader context for the unique aroma profile of RGT.
The proteome profiling section effectively maps out the impact of the green-making process on Rougui Tea aroma metabolism through a detailed analysis of differentially expressed proteins (DEPs). However, it could be enhanced by discussing the specific roles and functions of key DEPs in aroma formation, providing insights into how these proteins contribute to the development of the tea's distinct flowery and fruity aroma. Additionally, while KEGG enrichment analysis identifies important metabolic pathways, elaborating on the interaction between these pathways and how they collectively influence aroma compound synthesis would offer a more holistic view of the proteomic changes. Addressing the connection between the identified proteins and the sensory attributes of the tea, such as specific aroma profiles, would also bridge the gap between the molecular findings and practical applications in tea flavor enhancement.
The analysis between transcriptome and proteome provides key insights into gene regulation and metabolic processes but could further explore the implications of these findings on organismal function and adaptation. Specifically, discussing how differential gene and protein expression impacts physiological traits or stress responses would enhance our understanding. Additionally, incorporating comparative analyses with other species or conditions could reveal evolutionary trends or adaptive strategies. Again, the potential applications of these findings in biotechnology or medicine, such as in crop improvement or disease treatment, remain unaddressed and could significantly broaden the impact of the study.
The analysis thoroughly investigates DEGs and DEPs related to aroma biosynthesis but could enhance understanding by addressing the functional impact of these expression changes on sensory qualities of RGT, exploring the potential regulatory mechanisms behind the observed expression patterns, and examining environmental or cultivation factors that might influence these biosynthetic pathways. Additionally, linking these findings to practical applications in tea breeding or processing for aroma enhancement would provide a broader context and applicability.
The analysis could benefit from investigating how protein-protein interactions change under various conditions, validating key network nodes beyond qRT-PCR, and connecting these networks to specific RGT traits for practical applications.
Conclusions
This section could be enhanced by addressing the potential impact of environmental factors on aroma profiles, exploring the sensory evaluation with a larger and more diverse panel to validate the findings, and discussing the implications of these results for tea cultivation and processing practices. Further research might also explore genetic engineering or breeding strategies to optimize aroma profiles based on these findings.

Comments on the Quality of English Language
Minor editing of English language required.
Author Response
1.Response: Thank you for your valuable comments, we have reduced the similarity index of the manuscript, mainly which is on methodology.
2.Response: Thank you for your valuable comments, We have added that part on Line 84-97.
3.Response: Thank you for your valuable comments, We have added how do these compounds interact to form the unique aroma of RGT on Line 272-277 and Line 286-291.
4.Response: Thank you for your valuable comments, we have added that on Line 387-396.
5.Response: Thank you for your valuable comments, we have added that on Line 486-489.
6.Response: Thank you for your valuable comments, we have added that on Line 684-689.
7.Response: Thank you for your valuable comments, we have added that on Line 714-720.
8.Response: Thank you for your valuable comments, we have added that on Line 755-761.
Reviewer 2 Report
Comments and Suggestions for Authors
Dear Authors,
In general, the manuscript is good to read, the structure of the work is clear. The authors performed a lot of determinations and analyses. Statistical methods are satisfactory.
1. The general note applies to the entire manuscript. I ask authors to check and put spaces. For example: line 22 before the word "Mechanical" or line 27 before the word "This", etc...
2. Figures: 1, 2, 3 and 4. The font of the descriptions of the X and Y axes and some names of the considered cases should be increased, as in the case of Figure 1A (CK1R-1, CK1R-2 etc...)
3. Lines 449 and 450. The description for the drawing should be moved below the figure. Similarly Figure 5.
4. The keywords need to be modified, for example: “Wuyi Rock Tea, transcriptomes; proteomics; volatile organic compounds; green-making process; gas chromatography”.
5. Line 84. I ask the authors to consider changing the sentence: "In this study,..." to "The aim of the research was to use transcriptome, proteome...".
6. In the section: "2. Materials and Methods" there should be spaces between the following sentences (subsections): "2.1. Sample preparation and collection"; "2.2. Analysis of aroma components" etc...
7. Lines 214 - 217. These sentences are not needed in the Results section. The authors wrote about it in the "Materials and methods" section.
8. Can the authors present in the results what relationships are described by each main component, PC1 and PC2 (Figure 1A)? Is such an interpretation of the results possible?
9. Subsection: "3.3. Proteome profiling of proteins differentially expressed in response to the green-making process" is poorly discussed with the literature.
Author Response
1.Response: Thank you for your valuable comments, we have added the spaces in the whole manuscript.
2.Response: Thank you for your valuable comments, we have revised the group description in the diagram.
3.Response: Thank you for your valuable comments, we have revised them.
4.Response: Thank you for your valuable comments, we have revised the keywords.
5.Response: Thank you for your valuable comments, we have revised the sentence.
6.Response: Thank you for your valuable comments, we have revised them in the whole manuscript.
7.Response: Thank you for your valuable comments, we have deleted.
8.Response: Thank you for your valuable comments, we have modified the diagram.
9.Response: Thank you for your valuable comments, we have added the dicussion on Line 387-396.
Reviewer 3 Report
Comments and Suggestions for Authors
General comment
The manuscript ''Effect of mechanical damage in green-making process on aroma of Rougui Tea'' studies the process of green-making and it influence on aroma of Rougui Tea. After drying, samples were analyzed by GC-TOF-MS. Resulst showed that mechanical injury and dehydration could activate the up-regulated expression of genes related to the formation pathways of aroma, but the regulation of protein expression was not completely consistent. The long-term mechanical injury and dehydration could significantly increase the terpene terpenoids and esters compounds. . Transcriptome data showed that green-making and spreading process down-regulated a large number of genes in fresh leaves of RGT. Interestingly, the gene expression pattern during green-making and spreading process was similar.
Minor comments.
Abstract
Explain all abbreviation (AOS, MVA), not only few.
Line 12: What is Rougui Tea exactly? Add Latin name of species.
Line 13: add pace before The
Line 19: add pace before Transcriptomic... (check and corect through all Abstract)
1. Introduction
Line 32: Describe what is Wuyi Rock Tea (WRT), Rougui Tea (RGT) exactly.
2. Materials and Methods
Line 94: fresh leaves of which plant species? Please, add. Also, add GPS coordinates and elavation of collection site. It is ordinary for scientific works.
Line 110, 171 (and through all manuscrit): change short hyphnen into symbol for minus
Line 168: add pace before mL
Line 203: font size in μL?
Line 168: add pace before (
3. Results
Line 218: Table S3 should be incorporated in manuscript as Table 1. This is one of the most interesting findings.
Figure 4: rearrange title
Figure 5: add title below the figure (not above)
Line 627, 633: font size?
References
The references are not always arranged according to the journal style. Please, check. For example, put name of species in Italic (see lines 708, 709, 759....) and add dot in some journal abbreviations.
Author Response
1.Response: Thank you for your valuable comments, we have added explanations in the abstract.
2.Response: Thank you for your valuable comments, Wuyi Rock Tea (WRT), produced in the Wuyi Mountains of northern Fujian Province in China, is widely acknowledged as one of the most distinctive varieties of oolong tea for its exceptional “rock flavor”. Among the various cultivars of WRT, Rougui (RGT) and Shuixian (SXT) are the most widely grown ones.
3.Response: Thank you for your valuable comments, we have added the spaces.
4.Response: Thank you for your valuable comments, we have added the spaces of the whole manuscript.
5.Response: Thank you for your valuable comments, we have added the describe about Wuyi Rock Tea (WRT) and Rougui Tea (RGT) on Line 34-35.
6.Response: Thank you for your valuable comments, we have added on Line 113-114.
7.Response: Thank you for your valuable comments, we have revised.
8.Response: Thank you for your valuable comments, we have revised.
9.Response: Thank you for your valuable comments, we have revised.
10.Response: Thank you for your valuable comments, we have added.
11.Response: Thank you for your valuable comments, Table S1 is q-PCR reaction system, Table S2 is Real-time PCR gene primer sequence, and Table S3 is the aroma components of tea samples.
12.Response: Thank you for your valuable comments, we have rearranged.
13.Response: Thank you for your valuable comments, we have revised.
14.Response: Thank you for your valuable comments, we have revised.
15.Response: Thank you for your valuable comments, we have revised.
Round 2
Reviewer 1 Report
Comments and Suggestions for Authors
The authors meticulously completed the necessary revisions. Manuscript acceptable for publication.
Comments on the Quality of English Language
Minor editing of English language required.